# Investigation of the Influence of PLA Molecular and Supramolecular Structure on the Kinetics of Thermal-Supported Hydrolytic Degradation of Wet Spinning Fibres

**DOI:** 10.3390/ma13092111

**Published:** 2020-05-02

**Authors:** Małgorzata Giełdowska, Michał Puchalski, Grzegorz Szparaga, Izabella Krucińska

**Affiliations:** Centre of Advanced Technologies of Human-Friendly Textiles ‘Pro Humano tex’, Institute of Material Science of Textiles and Polymer Composites, Lodz University of Technology, Żeromskiego 116, 90-924 Łódź, Poland; malgorzata.gieldowska@dokt.p.lodz.pl (M.G.); grzegorz.szparaga@p.lodz.pl (G.S.); izabella.krucinska@p.lodz.pl (I.K.)

**Keywords:** polylactide, thermal degradation, hydrolytic degradation, fibres, kinetic of erosion, kinetics of degradation

## Abstract

In this study, differences in the kinetics of the thermal-supported hydrolytic degradation of polylactide (PLA) wet spinning fibres due to material variance in the initial molecular and supramolecular structure were analysed. The investigation was carried out at the microstructural and molecular levels by using readily available methods such as scanning electron microscopy, mass erosion measurement and estimation of intrinsic viscosity. The results show a varying degree of influence of the initial structure on the degradation rate of the studied PLA fibres. The experiment shows that hydrolytic degradation at a temperature close to the cold crystallization temperature is, on a macroscopic level, definitely more rapid for the amorphous material, while on a molecular scale it is similar to a semi-crystalline material. Furthermore, for the adopted degradation temperature of 90 °C, a marginal influence of the pH of the degradation medium on the degradation kinetics was also demonstrated.

## 1. Introduction

Poly(lactic acid) or polylactide (PLA) is the most commonly used biodegradable material, produced from completely renewable sources such as sugar, corn or other vegetables [1]. This thermoplastic aliphatic polyester exhibits similar mechanical properties to popular petroleum-based polymers, with additional special properties such as compostability and biocompatibility/bioresorbability [2,3]. According to the physical and chemical properties, PLA is a promising alternative to petroleum-based polymers from an application point of view. PLA can be used to form foams [4,5], films [6], fibres [7,8,9] and nonwovens [10,11] designed for many different applications, from medical [12,13] to agricultural [14,15] use.

Commercially available PLA is synthesised by polycondensation of lactic acid (poly(lactic acid)) or ring-opening polymerisation of lactide obtained from the depolymerisation of oligomers of lactic acid (polylactide), which is a product of the fermentation of biomass, such as corn [16].

The physical and chemical properties of final PLA products depend on the chirality of the polymer chains, and on the different supramolecular structures of the polymer chains. The high chirality of PLA chains reduces its ability to create a crystalline phase, which has a strong influence on the useful properties of the final products [17,18]. 

PLA is also well known as a hydrolysable and unstable biodegradable polyester. The degradation of this aliphatic polyester depends on the physical, chemical and biological agents used and is an interesting subject from a scientific point of view. The degree and rate of degradation also depend on the molecular and supramolecular structures of the polymer. The current state of knowledge on PLA is based on experiments carried out under laboratory and natural conditions. The described results from analysing degradation under laboratory conditions are focused on the influence of the content of the d-lactide isomer [19], the crystalline form [20], the content of nanomaterials [21,22] and the pH and temperature of the degradation medium [23] on the rate of hydrolytic degradation, which is the main way for PLA degradation; however, the tests were carried out on model samples and applied a limited number of degradation factors. The results presented in these works testify that hydrolytic degradation of polymer structures lasts up to many weeks and that it is favourable to conduct it in conditions close to the glass transition temperature of the polymer [24,25,26]. Other investigations of PLA degradation carried out under laboratory conditions are thermal degradation [27,28], artificial weathering [29,30] and composting [31,32].

An interesting issue in the field of life cycle assessment of PLA materials is testing in real conditions, in which the true degradation time of PLA products can be verified. These tests should take into account climatic conditions and the environment, including soil composition, and the degradation time may even be several years depending on the structure of the initial material [33,34].

In this paper, the results of investigating the influence of the initial molecular and supramolecular structure of polylactide on the thermal-supported hydrolytic degradation of PLA wet-spinning fibres are presented. The experiment was carried out with fibres characterised by various molecular structures, including the molar mass and content of D-lactide isomer of the polymer, and various supramolecular structures, especially the degree of crystallisation and crystal form. Experiments carried out on textile objects provide extremely valuable knowledge regarding the development of biodegradable materials. Materials such as fibres are not only semicrystalline, but also have an overall orientation, which is obtained by complex processes such as the draw ratio. The process of degradation was carried out in a selected water base medium with pH 3.5, 5 and 10, under temperatures that did not greatly exceed the glass transition point and were near the average cold crystallisation point for the studied fibres, 90 °C [35]. The reason for conducting the degradation process at an elevated temperature was to reduce the time of the experiment and also to check how the temperature-induced thermal condition above the glass transition point of PLA affected the rate of hydrolytic degradation of oriented fibres by various draw ratios [36]. The detailed information about the fibres, their properties and the methodology of how they were made were presented earlier [35]. The performed experiment allows us to clarify how the initial molecular and supramolecular structure of PLA impacts the rate of thermal-supported hydrolytic degradation of real objects that are ready for practical application, such as wet-spinning fibres. The degradation progress was measured as mass loss, supplemented by photographic and scanning electron microscopy (SEM) documentation, and an analysis of the change in intrinsic viscosity as the parameter characterising the degradation on the molecular level. The obtained results were numerically analysed in order to determine the factors of kinetic degradation at the molecular and macroscopic levels, which is important data in the evaluation of the influence of the structure of the initial material on the rate of degradation in the example of real objects such as fibres.

## 2. Materials and Methods

### 2.1. Materials

An investigation of the influence of initial molecular and supramolecular structures of PLA on the rate of thermal-supported hydrolytic degradation was performed on wet-spinning fibres made from a commercially available polymer, PLA Ingeo (Nature Works LLC, Minnetonka, MN, USA). The method of fibre preforming and its physical properties were described in detail by Puchalski et al. 2017 [35]. Table 1 shows the main parameters of the applied polymer, including the NatureWorks symbol, content of D-lactide isomer, weight-average molar mass (M_w_), dispersity (M_w_/M_n_) and the structural parameters of the studied samples: total draw ratio during fibre processing, crystal form, degree of crystallinity (χ_c_) and linear mass. In this study, we decided to use the symbols of PLAXX-DRYYY fibre samples, where XX is the d-lactide content in the polymer used and YYY is the value of the fibres’ total draw ratio.

### 2.2. Methodology of Thermal-Supported Hydrolytic Degradation

The hydrolytic degradation process was carried out in three selected mediums based on distilled water with various pH: pH 10 (water solution of sodium carbonate), pH 5 and pH 3.5 (water solution of acetic acid). Samples of the same mass, 5 g, were degraded in 50 mL of hydrolytic medium under a controlled temperature of 90 °C for 1, 2, 3, 4, 5, 6, 7, 10, 14 and 21 days.

### 2.3. SEM Method

The effects of degradation on the change of PLA fibre morphology were studied by using photography documentation and a Nova NanoSEM 230 scanning electron microscope (SEM) from FEI Company (Eindhoven, The Netherlands). For the SEM measurement, the fibre samples were prepared by fixing the parts of fibres to an SEM holder using conducting carbon adhesive tape. The studies were carried out using a low-vacuum mode and beam energy of 10 keV, which eliminated the requirement to cover the sample with a conductive material such as gold.

### 2.4. Mass Loss

The measurement of the mass after the degradation of samples, which were cleaned with distilled water, was conducted using a PS.R1 precision balance (Radwag, Radom, Poland). The mass percent remaining after the time of degradation (D_t_) was calculated according to the following equation:(1)$$D_{t}=\frac{m_{t}}{m_{0}}100\%$$
where m_0_ and m_t_ are the masses of the sample before and after degradation, respectively.

### 2.5. Intrinsic Viscosity

Structural changes at the molecular level during thermal-supported hydrolytic degradation were estimated by determining the intrinsic viscosity of diluted polymer/dichloromethane (0.08 g/dL) using an Ubbelohde viscometer (Type 2a, Poland) at 25 °C. The relationship between the viscosity-average molecular weight (Mη) and estimated intrinsic viscosity [η] can be described by the following Mark–Houwink Equation [37]:(2)$$\left[\eta \right]={\mathrm{KM}}_{\eta }{}^{\alpha }$$
where K and α are constants, which for PLLA equal 1.124 × 10^–2^ and 0.52, respectively, and satisfactorily describe the tested PLA from Nature Works LLC with a slight content of d-lactide isomer [38]. Since the studied polymers changed at the molecular level during degradation, an analysis of the estimated intrinsic viscosity change as a function of the degradation time was performed.

## 3. Results and Discussion

### 3.1. Photographic Documentation and SEM Results

First, the changes of morphology of the PLA fibres after the thermal-supported hydrolytic degradation process were characterised. All of the investigated samples were completely degraded after 21 days. Figure 1 shows selected photographs of the degradation process of the studied samples occurring in various hydrolysed mediums. According to these photographs, the physical changes of samples during thermal-supported hydrolytic degradation depended on the initial ordering and crystallinity of the sample, and were visible as strong shrinkage after the first day of the experiment in the case of amorphous samples. The relationship between supramolecular ordering and the shrinkage phenomenon of the fibrous PLA structure was well known, as reported by, e.g., Puchalski et al. [39]. Shrinkage of fibrous materials resulted from the disordered pre-existing supramolecular structure of the polymer, which has a tendency towards relaxing and ordering, mainly during thermal processing. Therefore, the reason for their amorphous initial structure, the rapid and significant shrinkage of PLA12-DR400 and PLA12-DR600 fibres, was possible to predict. The opposite occurred with PLA1.4-DR500 and PLA1.4-DR650, in which, due to the initial semicrystalline structure, the shrinkage was marginal. A very interesting phenomenon was observed in the investigation of the thermal-supported degradation of fibres made from PLA with 2.5% d-lactide. The rapid shrinkage of PLA2.5-DR450 fibres occurring with a lower draw ratio was observed, while in the case of fibres with a higher draw ratio (PLA2.5-DR550), this phenomenon was insignificant. That result confirmed the relationship between shrinkage during thermal-supported hydrolytic degradation and the pre-existing supramolecular structure created in the technological regime. The next step of degradation was fragmentation, which was observed after the third and fifth days of degradation despite the ordering of the pre-existing supramolecular structure. According to the photographic documentation (Appendix A), the most degradable fibres were obtained from PLA with the highest D-lactide content and molar mass and an amorphous supramolecular ordering; after the third day, the samples were in powdered form, which was the effect of the first stage of the degradation process, fragmentation. During the degradation time, the volume of the samples decreased, suggesting a significant mass loss, the kinetics of which are presented in the next part of this paper. The most important information obtained from the photographic documentation is the lack of a clearly visible influence of the change of pH of the hydrolytic medium on the rate of degradation. Regarding the organoleptic evaluation of the degradation effects, it can only be concluded that in the case of thermal-supported hydrolytic degradation, the initial structure of the polymer and the regimes of its processing, which determine the ordering and crystallinity of the fibres, have a significant influence on the rate of degradation.

A more exhaustive analysis of the morphological structure changes of PLA fibres during degradation was obtained by using scanning electron microscopy. Figure 2 shows representative SEM images of samples before and after five days of thermal-supported hydrolytic degradation recorded at a magnification of ×2000. The initial structures of the investigated samples were different. The PLA12-DR400 and PLA12-DR600 fibres were characterised by the least textured surfaces, the surfaces of PLA2.5-DR450 and PLA2.5-DR550 fibres were wavy, while the surface texture of PLA1.4-DR500 and PLA1.4-DR650 was smooth but contained transverse elements occurring periodically. The SEM results after five days of degradation clearly showed the evolution of the morphology of the samples as the result of various mechanisms of thermal-supported degradation. All studied materials were fragmented after five days, which is the first stage of degradation of polymeric materials. The degradation of the studied materials was combined with the process of PLA disintegration and fragmentation, which takes place in the areas of erosion of the amorphous structure of the fibre material, which was described in detail by Azimi et al. [40]. This was confirmed by the presence of transverse cracks in the samples, especially in PLA12-DR400 and PLA12-DR600. The last degradation mechanism that was clearly visible was erosion, mainly surface erosion, as illustrated by the changes of surface texture of the studied samples. The analysis of the investigations by SEM showed differences in the degradation rate of the studied samples, varying by the initial polymer structure and supramolecular structure of the fibres. In the case of the PLA12-DR400 and PLA12-DR600 samples, the amorphous SEM results are proof of rapid fragmentation before surface erosion. In contrast, the SEM results verify the strong tendency of surface erosion in the semicrystalline samples PLA2.5-DR450 and PLA2.5-DR550, while the semicrystalline PLA1.4-DR500 and PLA1.4-DR650 samples mainly underwent fragmentation and erosion of the amorphous phase of materials.

### 3.2. Mass Loss Kinetics

The photographic documentation and SEM results clearly show the evolution of the samples’ morphology during degradation at the macroscopic and microscopic scales, and verify the shrinkage, fragmentation and erosion of the studied fibrous materials. The next step of the investigation was to analyse mass loss in the function of degradation time measured as the mass percent remaining, according to Equation (1). Figure 3 shows the changes of the mass percent remaining (D_t_) of the studied samples during thermal-supported hydrolytic degradation in a selected water solution with various pH levels. A significant decrease of the mass percent remaining for the samples obtained from PLA containing 2.5% and 12% D-lactide was observed after the second day, while for fibres obtained from PLA with 1.4% D-lactide it was only after the fifth day.

Table 2 shows estimations of the two characteristic kinetic parameters describing the typical erosion profile, onset time (t_on_) and observed pseudo-first-order rate of erosion constant (k_e_). The latter was characterised as a slope value according to the following equation [41]:(3)$$\ln\left(D_{t}\right)=A-k_{e}t$$
where D_t_ is the mass percent remaining after the time of degradation, calculated according to Equation (1), t is the time of degradation starting from when erosion was significant, and A is an intercept. The calculated t_on_ value is derived from intersecting the regression line in Equation (3) with the initial mass value as follows:(4)$$t_{\mathrm{on}}=\frac{A-\ln\left(100\right)}{k_{e}}$$

Considering the limited data points for each investigation, Table 2 shows good adherence to Equation (3), with a high correlation coefficient (R) and reasonably small relative standard error (SE) in k_e_, and with intercept values estimated by the use of OriginPro software (version 8.6, OriginLab Corporation, Northampton, MA, USA). The analysis of the estimated kinetic parameters of the erosion profiles of PLA fibres degraded by thermal-supported hydrolysis clearly presents the influence of the initial PLA molecular structure on the pseudo-first-order rate constant and onset time values. The k_e_ increased with an increased content of D-lactide isomer and decreased weight-average molar mass, as expected. It is worth noting that it is difficult to define the influence of the crystallinity degree of the investigated samples on the pseudo-first-order rate of erosion constant value. In contrast, it is in the case of t_on_ where both the molecular structure of the initial polymer and the supramolecular structure of fibres influence this parameter. The onset time increased with an increased content of D-lactide, weight-average molar mass and crystallinity degree. The highest pseudo-first-order rate of erosion constant, around 0.3 days^–1^, and lowest onset time, around 0.35 days, characterised the amorphous PLA12-DR400 and PLA12-DR600 fibres, while for the PLA1.4-DR500 and PLA1.4-DR650 fibres the maximum k_e_ was 0.095 days^–1^ and 0.087 days^–1^ and t_on_ was 1.698 days and 2.116 days, respectively. Thus, the experiment also showed that the pre-existing ordering and crystalline structure influenced the kinetics of PLA degradation in the proposed conditions, and the shortest onset time was observed for fibres characterised by an amorphous structure (less than one day) and the longest onset time for materials with a semicrystalline structure of ordered α form (more than two days).

### 3.3. Degradation Kinetics of PLA Fibres on Molecular Level

The degradation of polymers is mainly investigated on the molecular level by means of size exclusion chromatography (SEC) [41] or gel permeation chromatography (GPC) [42], by which it is possible to analyse the number average molar mass (M_n_), weight-average molar mass (M_w_) and dispersity (M_w_/M_n_). In our experiment, according to the possibilities of investigation, we decided to analyse the changes of fibres at the molecular level by means of a viscometer, which, according to Equation (2), allows that analysis to be performed.

The analysis of changes of the measured intrinsic viscosity of PLA fibres during thermal-supported hydrolytic degradation was carried out up to the first seven days, resulting from the strong mass loss after seven days (described above) and finally in the impossibility of preparing experimental samples. Figure 4 shows the relative changes of [η] due to the medium with various pH levels. All studied samples indicated a change in the molecular level after the first day, but the most intense was in the material formed from PLA containing 12% D-lactide isomer and characterised by the highest molar mass.

To insightfully analyse the thermal-supported degradation rate of the studied samples, the degradation rate constant (k_d_) was calculated from the decreased relative intrinsic viscosity based on the first-order kinetic model according to the following equation [43]:(5)$$\ln\left(\frac{{[\eta }_{t}]\;}{{[\eta }_{0}]}\right){=A-\alpha k}_{d}t$$
where [η_t_]/[η_0_] is the percentage change of the intrinsic viscosity after time degradation [η_t_] due to the initial intrinsic viscosity [η_0_], α is a constant according to Equation (2), t is the time of degradation and A is an intercept.

According to the apparent degradation rate, the degradation time of the half intrinsic viscosity is calculated by the following equation:(6)$$t_{50\%}=\frac{A-\ln\left(50\right)}{{\alpha k}_{d}}$$

The presented first-order kinetic model was used to describe the thermal-supported degradation of the studied fibres based on the estimated intrinsic viscosity. Table 3 shows the kinetic parameters of the investigated degradation.

Similar to the mass loss kinetics analysis, Table 2 shows good adherence to Equation (5), with a high correlation coefficient (R) and relatively small relative standard error (SE) in k_d_, and with intercept values estimated by OriginPro 8.6 to assess the relative intrinsic viscosity change kinetics. However, the performed experiment clearly showed the various characteristics of degradation of the molecular structure of PLA more than the macroscopic mass erosion of the samples.

With regard to degradation at the molecular level, it is difficult to unequivocally find the influence of the initial polymer structure on the degradation kinetics. The estimated degradation rate values constantly decrease insignificantly with a decreased content of D-lactide isomer and Mw and an increased crystallinity degree. The influence of the initial structure on the kinetics of degradation is more pronounced for t_50%_ when it increases with a decreased content of D-lactide isomer and weight-average molar mass, and also with an increased crystallization degree. It is worth noting that the degradation time of the half intrinsic viscosity was less than two days for all the studied samples, and the highest was for the PLA1.4-DR650 sample, in which the crystalline α form was detected. The pH of the environment also has a slight influence on the kinetic factors of the thermal-supported hydrolytic degradation of PLA fibres at the molecular level, but it is not possible to predict which pH value will be more favourable. For the most amorphous sample, the lower value of t_50%_ was estimated for pH 5, but for the most crystalline sample with a α form crystal it was pH 3.5.

## 4. Conclusions

The main goal of this investigation was to present differences in the kinetics of the hydrolytic degradation of PLA due to real material variance in the molecular and supramolecular structure of sample wet-spinning fibres. Complementary studies were realised on various levels by using selected methods such as SEM and viscosimetry.

The thermal-supported hydrolytic degradation experiment allows us to demonstrate the influences of temperature, or heat transfer, on the kinetics of hydrolytic degradation. In the sample erosion, decreasing the molar mass (intrinsic viscosity) was significantly more rapid than at lower temperatures, the results of which are described in the cited literature.

From a macro- and microscopic point of view, all of the studied fibres became fragmented after just three days, and surface and volume erosion were observed using SEM. The initial structure of the studied biodegradable fibres had a strong effect on the degradation, which was shown by photographic documentation and analysis of mass loss kinetics. The onset time for the amorphous material with a high molar mass and D-lactide content was less than one day, while for the semicrystalline material it was nearly two days. The erosion profiles and pseudo-first-order rate of erosion constant were also variable and dependent on the initial structure. It is worth noting that the experiment demonstrated the lack of influence or insignificant influence of the pH of the applied degradation medium on the process kinetics. This is due to an increase in temperature of the process to a value where the pH is not affected by hydrolysis.

The molecular structure was also changed during thermal-supported hydrolytic degradation, but in this case the influence of the initial structure on the process rate was less significant than in the macrostructural change. The calculated degradation time of the half intrinsic viscosity for the amorphous samples with the highest molar mass was around one day, while for the semicrystalline material with the lower molar mass it was around 1.5 days. Based on the obtained results, it is supposed that the initial supramolecular structure has an effect on the degradation rate at the molecular level.

To summarize the experiment, it should be stated that the initial structure and supramolecular ordering had the greatest influence on the macroscopic effects of the degradation. In the case of the objects tested, the kinetics of changes depended not only on the molecular structures of the PLA used but also on the draw ratio applied; therefore, in the case of fibres, the overall orientation influences the kinetics of degradation, which will be further investigated in detail. In addition, for the adopted degradation temperature of 90 °C, the influence of the pH of the degradation medium on the degradation kinetics was marginal, which could be expected in the case of rapid degradation leading to the disintegration of the polymer and a subsequent change in the pH of the medium, which will also be examined in more detail in future studies.

## Figures and Tables

**Figure 1 materials-13-02111-f001:**
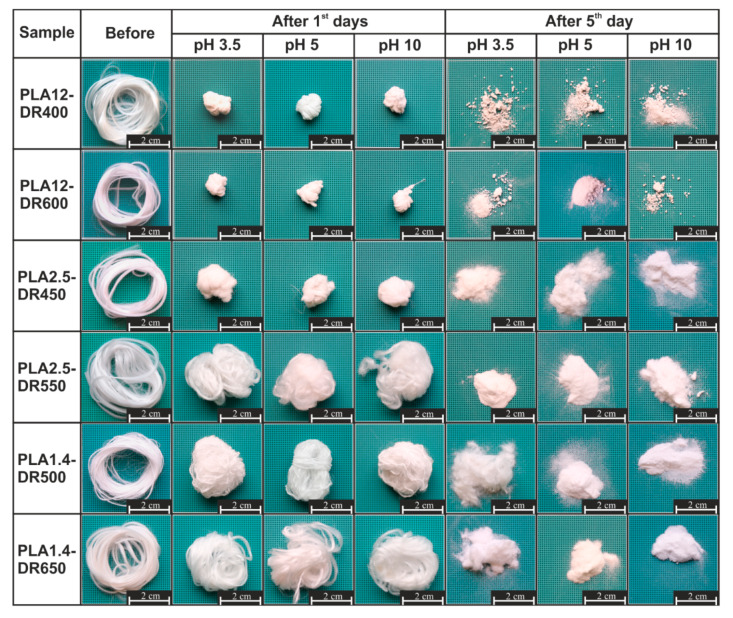
Photographic documentation of the thermal-supported hydrolytic degradation of fibres obtained from PLA.

**Figure 2 materials-13-02111-f002:**
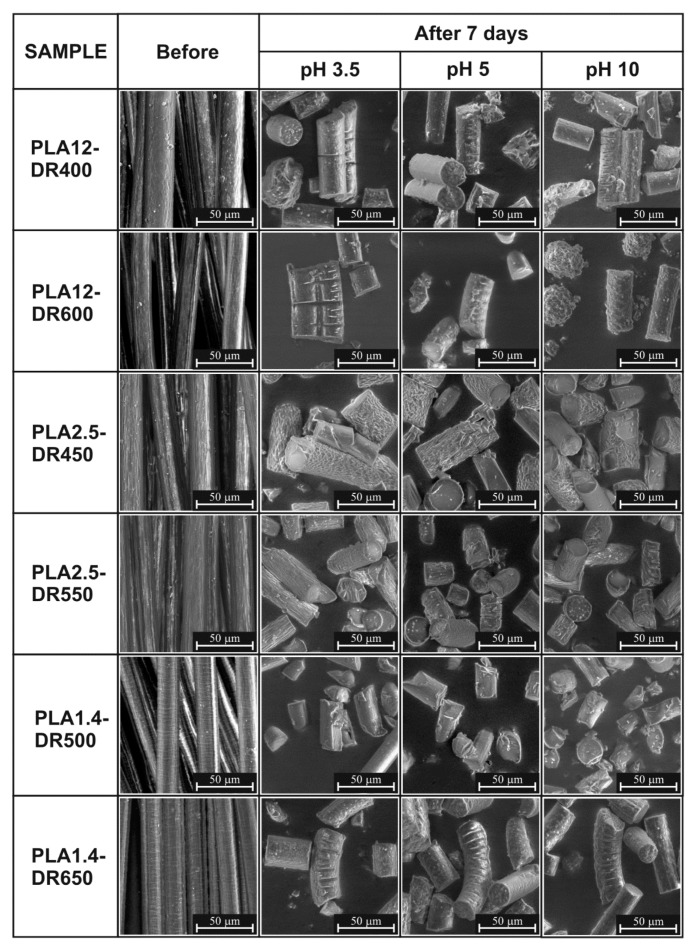
SEM results of the studied samples recorded before and after seven days of thermal-supported hydrolytic degradation.

**Figure 3 materials-13-02111-f003:**
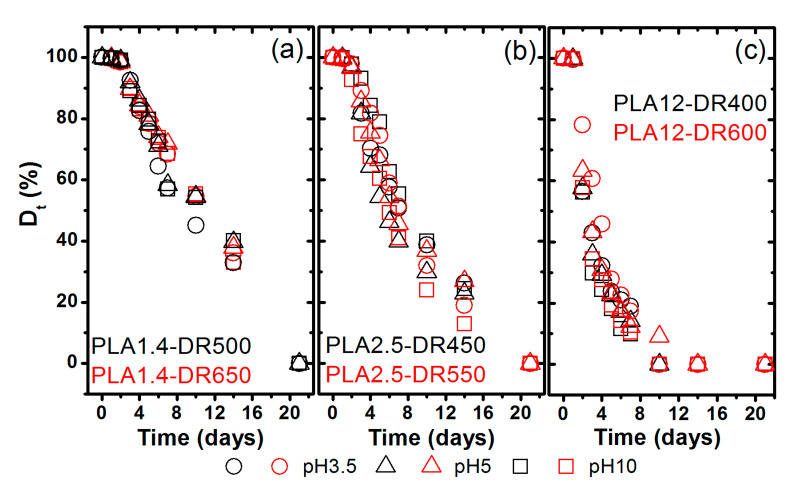
Changes of the mass percent remaining of the studied samples during thermal-supported hydrolytic degradation: (**a**) fibres made from PLA Ingeo 6201D, (**b**) fibres made from PLA Ingeo 2002D, and (**c**) fibres made from PLA Ingeo 4060D.

**Figure 4 materials-13-02111-f004:**
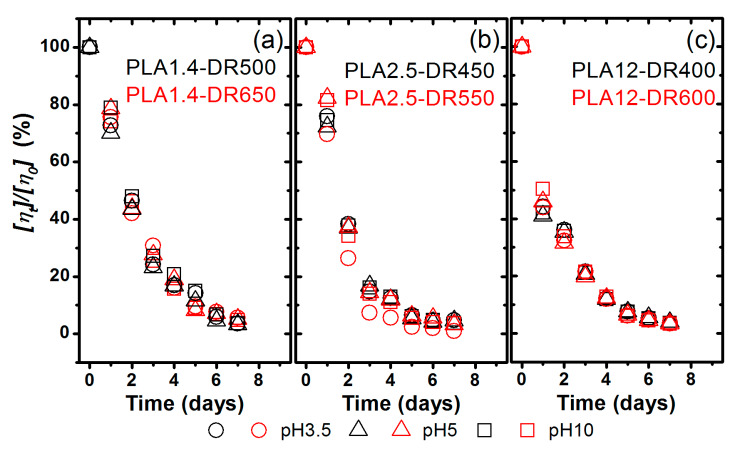
Changes of the intrinsic viscosity percent remaining of the studied samples during thermal-supported hydrolytic degradation: (**a**) fibres made from PLA Ingeo 6201D, (**b**) fibres made from PLA Ingeo 2002D, and (**c)** fibres made from PLA Ingeo 4060D.

**Table 1 materials-13-02111-t001:** Main characteristics of the studied fibres and raw materials described in detail by Puchalski et al. 2017 [35].

Sample	Characteristic of Raw Polymer				Characteristic of Fibre			
	Nature Works Symbol of PLA	Contents of d-Lactide Isomer (%)	M_w_ (kg/mol)	M_w_/M_n_	Total Draw Ratio (%)	Crystal Form *	χ_c_ ** (%)	Linear Mass (tex)
PLA12-DR400	Ingeo 4060D	12	119	1.40	400	−	amorphous	158.00 (2.09 ***)
PLA12-DR600	Ingeo 4060D	12	119	1.40	600	α′	1.2	80.33 (0.90)
PLA2.5-DR450	Ingeo 2002D	2.5	112.6	1.46	450	α′	16.6	121.00 (1.43)
PLA2.5-DR550	Ingeo 2002D	2.5	112.6	1.46	550	α′	33.5	72.67 (0.79)
PLA1.4-DR500	Ingeo 6201D	1.4	59.1	1.29	500	α′	47.6	96.00 (1.04)
PLA1.4-DR650	Ingeo 6201D	1.4	59.1	1.29	650	A	53.8	68.33 (0.52)

* Parameter determined by using the WAXD method. ** Value estimated by using the DSC method. *** Coefficient of variation is in brackets.

**Table 2 materials-13-02111-t002:** Kinetic parameters of the mass percent remaining of the studied samples during thermal-supported hydrolytic degradation.

Sample	pH of Medium	A ± SE	k_e_ ± SE (Days^−1^)	R	t_on_ (Days)
PLA12-DR400	3.5	4.65 ± 0.13	0.29 ± 0.03	0.962	0.33
PLA12-DR400	5	4.71 ± 0.12	0.31 ± 0.03	0.959	0.34
PLA12-DR400	10	4.78 ± 0.14	0.38 ± 0.03	0.980	0.43
PLA12-DR600	3.5	4.86 ± 0.06	0.30 ± 0.01	0.988	0.83
PLA12-DR600	5	4.85 ± 0.05	0.34 ± 0.01	0.993	0.72
PLA12-DR600	10	4.80 ± 0.10	0.36 ± 0.02	0.988	0.69
PLA2.5-DR450	3.5	4.73 ± 0.06	0.11 ± 0.04	0.989	1.12
PLA2.5-DR450	5	4.74 ± 0.09	0.12 ± 0.01	0.974	1.15
PLA2.5-DR450	10	4.76 ± 0.05	0.12 ± 0.01	0.990	1.15
PLA2.5-DR550	3.5	4.78 ± 0.04	0.13 ± 0.03	0.992	1.33
PLA2.5-DR550	5	4.77 ± 0.06	0.11 ± 0.01	0.973	1.39
PLA2.5-DR550	10	4.81 ± 0.04	0.15 ± 0.04	0.997	1.35
PLA1.4-DR500	3.5	4.78 ± 0.03	0.10 ± 0.01	0.988	1.70
PLA1.4-DR500	5	4.73 ± 0.04	0.08 ± 0.01	0.969	1.63
PLA1.4-DR500	10	4.72 ± 0.05	0.08 ± 0.01	0.972	1.64
PLA1.4-DR650	3.5	4.77 ± 0.02	0.08 ± 0.01	0.992	2.01
PLA1.4-DR650	5	4.76 ± 0.02	0.08 ± 0.01	0.988	2.01
PLA1.4-DR650	10	4.79 ± 0.04	0.09 ± 0.01	0.976	2.12

**Table 3 materials-13-02111-t003:** Kinetic parameters of the intrinsic viscosity percent remaining of the studied samples during thermal-supported hydrolytic degradation.

Sample	pH of Medium	A ± SE	k_d_ ± SE (Days^−1^)	R	t_50%_ (Days)
PLA12-DR400	3.5	4.60 ± 0.16	0.69 ± 0.04	0.970	1.09
PLA12-DR400	5	4.57 ± 0.18	0.70 ± 0.04	0.962	1.09
PLA12-DR400	10	4.59 ± 0.17	0.70 ± 0.04	0.959	1.19
PLA12-DR600	3.5	4.44 ± 0.09	0.65 ± 0.02	0.985	1.35
PLA12-DR600	5	4.44 ± 0.08	0.65 ± 0.02	0.989	1.29
PLA12-DR600	10	4.50 ± 0.08	0.67 ± 0.02	0.988	1.31
PLA2.5-DR450	3.5	4.57 ± 0.19	0.66 ± 0.04	0.954	1.15
PLA2.5-DR450	5	4.56 ± 0.19	0.68 ± 0.05	0.954	1.06
PLA2.5-DR450	10	4.59 ± 0.14	0.68 ± 0.03	0.969	1.10
PLA2.5-DR550	3.5	4.45 ± 0.08	0.63 ± 0.02	0.989	1.33
PLA2.5-DR550	5	4.39 ± 0.10	0.61 ± 0.02	0.980	1.29
PLA2.5-DR550	10	4.43 ± 0.09	0.63 ± 0.02	0.986	1.34
PLA1.4-DR500	3.5	4.71 ± 0.10	0.64 ± 0.02	0.982	1.53
PLA1.4-DR500	5	4.68 ± 0.09	0.66 ± 0.02	0.987	1.57
PLA1.4-DR500	10	4.77 ± 0.11	0.63 ± 0.03	0.978	1.54
PLA1.4-DR650	3.5	4.66 ± 0.07	0.62 ± 0.02	0.990	1.70
PLA1.4-DR650	5	4.68 ± 0.10	0.61 ± 0.03	0.978	1.67
PLA1.4-DR650	10	4.64 ± 0.08	0.59 ± 0.02	0.985	1.72

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
