# Peer review of "Investigation of the Influence of PLA Molecular and Supramolecular Structure on the Kinetics of Thermal-Supported Hydrolytic Degradation of Wet Spinning Fibres"

_materials, 2020, doi:10.3390/ma13092111_

Round 1

Reviewer 1 Report

The manuscript ”Investigation of the influence of PLA molecular and supramolecular structure on the kinetics of thermal-supported hydrolytic degradation of wet spinning fibres” contributes with a study of the accelerated hydrolysis of different types wet spun PLA fibres. The following comments and questions need to be addressed before publication:

  1. The title of the manuscript is quite long and difficult to understand. What do the authors mean by molecular and supramolecular structure? This is true for the complete manuscript.
  2. It is commonly adapted that PLA is called poly(lactic acid) if it is made via step-growth polymerization and polylactide if it is made by ring-opening polymerization. The authors use both. Please choose the correct description.
  3. In the abstract, the authors state “The experiment shows that hydrolytic degradation at a temperature close to the cold crystallization temperature on a macroscopic level is definitely more rapid for the amorphous material” which becomes unfortunate as an amorphous material would not have a cold crystallization. Please rephrase. Also include a description of how the cold crystallisation temperature was determined for these samples.
  4. Page 1, line 30. Please specify what is meant by a “popular petroleum-based polyesters”.
  5. Page 2, line 51. As there has been extensive amount of research on PLA degradation and the influence of many combined factors effecting the degradation process and rate, I strongly urge the authors to rewrite the section including “the tests were carried out on model samples and applied a limited number of degradation factors.” This is simply not true. The authors are also citing quite recent work and should include also the early pioneering work in the field.
  6. In general, the introduction is stating the same thing many times and this should be corrected. There is also no aim given (it can be found in the conclusions), but a summary of the study is presented at the end of the introduction. Please remove this summary and include a clear aim to the study specifying what new information the study was aiming to contribute with. It is presently not clear what the novelty of the manuscript is.
  7. Please include a clear description of the naming system of the samples and also use one naming system.
  8. The conclusions drawn from the photographs and from the SEM images are slightly exaggerated. As it the amorphous phases that degrade first, what is the difference between the semi crystalline fibres and the amorphous fibres erosion behaviour? It is well known that the amorphous parts degrade first. Did the authors measure crystallinity as a function of time. This should be included.
  9. Page 3 line 190. I think the authors mean something else in this statement “A significant increase of mass percent remaining for the samples obtained from PLA containing 2.5% and 12% D-lactide was observed after the second day, while for fibres obtained from PLA with 1.4% D-lactide it was only after the fifth day.” Increase should most likely read decrease.
  10. Table 2. The number of significant digits seems large.
  11. It is difficult to understand the novelty of the study. That there are limited differences between the various PLA samples at such an elevated temperature is hardly surprising under the conditions used. That the effect of temperature will be predominant when PLA is placed in water at 90 C is expected. How did the pH change as a function of time. It is likely that the pH was similar in all samples very early on in the study. Please include the change in pH as a function of time.

Author Response

The answer on Reviewers Comments

We would like to express our grateful for Reviewer for their work, worth remarks and comments. Hereby we would like to answer to him:

  1. The title of the manuscript is quite long and difficult to understand. What do the authors mean by molecular and supramolecular structure? This is true for the complete manuscript.

The authors admit that the title is long, but presents the content of the article in a comprehensive way. The authors would like to point out that they studied the influence of differentiation of initial materials at the molecular level, i.e. different molar mass, dispersion, application of PLA with different D-lactide content, and different at the supramolecular level, i.e. different degree of crystallinity and crystalline form. Introduction to the title of information about different D-lactide content, different crystallinity might be more readable, but not full of suspicion as we mentioned additionally we have alpha and alpha prim crystalline structures and different molar mass. We count on your forbearance to keep the title without significant changes.

  1. It is commonly adapted that PLA is called poly(lactic acid) if it is made via step-growth polymerization and polylactide if it is made by ring-opening polymerization. The authors use both. Please choose the correct description.

Thank you for the remark. We have completely checked the manuscript and made corrections e.g. line 14.

  1. In the abstract, the authors state “The experiment shows that hydrolytic degradation at a temperature close to the cold crystallization temperature on a macroscopic level is definitely more rapid for the amorphous material” which becomes unfortunate as an amorphous material would not have a cold crystallization. Please rephrase. Also include a description of how the cold crystallisation temperature was determined for these samples.

We added corrected information and it is true for PLA4060 (amorphous) the cold crystallization as well as melting point is not observed but please see ref [35] (our previously work) where we show the insignificant crystallization of amorphous PLA during fibers made processing by wet spinning method.

  1. Page 1, line 30. Please specify what is meant by a “popular petroleum-based polyesters”.

Thank you for the remark. We change “polyesters” to “polymers”

  1. Page 2, line 51. As there has been extensive amount of research on PLA degradation and the influence of many combined factors effecting the degradation process and rate, I strongly urge the authors to rewrite the section including “the tests were carried out on model samples and applied a limited number of degradation factors.” This is simply not true. The authors are also citing quite recent work and should include also the early pioneering work in the field.

The reviewer is undoubtedly right that there are many papers on PLA degradation. Nevertheless, the reviewer should note that there are not many publications on the degradation of textile structures. It is possible that the intentions of the authors were unfortunately misinterpreted, but the truth is that most of the papers are studies of films and dog-bone shape materials. In one publication the influence of D-lactide content on degradation is analysed, in another pH of the medium. The authors did not find a publication similar to the submitted for review. The authors have additional results of WAXD and DSC, but they decided that it will be very interesting to show how the conditions of fiber formation process and polymer selection influence the degradation of the final product. If the reviewer thinks that we have omitted fundamental papers, please give them for citation.

  1. In general, the introduction is stating the same thing many times and this should be corrected. There is also no aim given (it can be found in the conclusions), but a summary of the study is presented at the end of the introduction. Please remove this summary and include a clear aim to the study specifying what new information the study was aiming to contribute with. It is presently not clear what the novelty of the manuscript is.

The relevant sentences have been added to the introduction and conclusion.

  1. Please include a clear description of the naming system of the samples and also use one naming system.

Thank you for the remark we add information in the section 2.1. Materials and additionally we rewritten sample codes in section 3.1. Photographic documentation and SEM results.

  1. The conclusions drawn from the photographs and from the SEM images are slightly exaggerated. As it the amorphous phases that degrade first, what is the difference between the semi crystalline fibres and the amorphous fibres erosion behaviour? It is well known that the amorphous parts degrade first. Did the authors measure crystallinity as a function of time. This should be included.

The authors have carefully analyzed the changes of crystallinity and general orientation, but will be the subject of another article, where DSC and WAXD studies will be presented. The results are very interesting, but they are too much to be included in this article. Introducing only partial DSC or WAXD results in the opinion of the authors would not be justified because it would only raise more questions that we would like to answer in the next article where we will analyze crystallizations during degradation.

  1. Page 3 line 190. I think the authors mean something else in this statement “A significant increase of mass percent remaining for the samples obtained from PLA containing 2.5% and 12% D-lactide was observed after the second day, while for fibres obtained from PLA with 1.4% D-lactide it was only after the fifth day.” Increase should most likely read decrease.

Thank you for the remark. We change this mistake.

  1. Table 2. The number of significant digits seems large.

We decided to round up numbers as was possible.

  1. It is difficult to understand the novelty of the study. That there are limited differences between the various PLA samples at such an elevated temperature is hardly surprising under the conditions used. That the effect of temperature will be predominant when PLA is placed in water at 90 C is expected. How did the pH change as a function of time. It is likely that the pH was similar in all samples very early on in the study. Please include the change in pH as a function of time.

The authors regret, but do not have a full analysis of pH changes as a function of time for this experiment. In fact, the results of our second experiment conducted in parallel showed that pH 10 as a function of degradation time is falling, but pH 5 and pH 3.5 are still without significant changes.

Reviewer 2 Report

Review of the manuscript “Investigation of the influence of PLA molecular and supramolecular structure on the kinetics of thermal supported hydrolytic degradation of wet spinning fibres” by Gieldowska, Puchalski, Szparaga, Krucinska.

This paper studies the hydrolytic degradation carried out at three different pHs and 90 ºC, of PLA fibers, obtained by wet spinning. The fibers were prepared in a previous work and differ in D-isomer content, molar mass and crystallinity, and were obtained with two different draw ratios. The goal was to study the effect of these molecular and supramolecular characteristics of the fibers on the morphology, mass loss kinetics and kinetics of degradation at a molecular level (measured by change in intrinsic viscosity). The authors find that the effect of these molecular and supramolecular characteristics is higher at a macroscopic level than at molecular scale, being the fiber with higher D-isomer content, higher molar mass and lower crystallinity the one with highest degradation rate.

I think that the subject of the paper is of interest and the experimental work is good, however, the manuscript is not well written Therefore, in my opinion the paper could be published but after minor revision addressing some points that I explain in the following:

  1. Major amendments:

#1. The manuscript needs a language revision; there are many sentences that are difficult to understand. In particular, Result and Discussion Sections need to be polished.

#2. References section should be carefully revised as it contains several errors:

  • At line 424 Reference [40] should be removed because it does not correspond with what is written at Line 172: “which was described in detail by Azimi et al. [40].”
  • In addition, the numbers of subsequent references should be corrected,
  • And references 45, 46 and 47 should be removed because they are not cited in the text.

.

  1. . Minor amendments

#3. In Table 1: Units of Linear Mass are missing.

#4. In Table 2 and Table 3: in the column “pH of medium” I think that the word “pH” written in each line should be removed because it is redundant as it appears at the head of the column.

#5. At line 229: it is written “3.2. Degradation kinetics of PLA fibres on molecular level” and it should be written 3.3. Degradation kinetics of PLA fibres on molecular level”

#6. In Supplementary file: Figure S3: the foot should be added.

Author Response

The answer on Reviewers Comments

We would like to express our grateful for Reviewer for their work, worth remarks and comments. Hereby we would like to answer to him:

Major amendments:

#1. The manuscript needs a language revision; there are many sentences that are difficult to understand. In particular, Result and Discussion Sections need to be polished.

The authors know very well that they are not grammar experts so the work was corrected by MDPI English Editors. However, if the reviewer sustain his comments, we will submit a complaint to MDPI English Editors. However, we have made some corrections with MDPI English Editors.

#2. References section should be carefully revised as it contains several errors:

At line 424 Reference [40] should be removed because it does not correspond with what is written at Line 172: “which was described in detail by Azimi et al. [40].”

We changed it.

In addition, the numbers of subsequent references should be corrected,

And references 45, 46 and 47 should be removed because they are not cited in the text.

We removed references.

Minor amendments

#3. In Table 1: Units of Linear Mass are missing.

We added unit.

#4. In Table 2 and Table 3: in the column “pH of medium” I think that the word “pH” written in each line should be removed because it is redundant as it appears at the head of the column.

We changed it.

#5. At line 229: it is written “3.2. Degradation kinetics of PLA fibres on molecular level” and it should be written “3.3. Degradation kinetics of PLA fibres on molecular level”

We changed it.

#6. In Supplementary file: Figure S3: the foot should be added.

We added footnote.

Reviewer 3 Report

The review article deals with very interesting issues related to the impact of the structure of polymer materials on the process of their aging resistance. The language page of the article is correct.

From the formal comments regarding the article, I would like to draw attention to the lack of consistency in the naming of samples. Once the samples are described as PLAX e.g. PLA12 and another time as PLAX-YYY e.g. PLA12-DR600.

From the substantive comments regarding the article, I would like to draw attention to:

  • Why is PLA12-DR400 material considered amorphous in the table, and is the crystallinity value already given in PLA12_DR600? Material with a degree of crystallinity of 1.2% would also be considered as amorphous, because with such a small value, the possible measurement error used to determine the degree of crystallinity raises doubts.
  • However, I have the biggest complaint about the lack of thorough research and description of the methodology of testing the degree of crystallinity of materials.

Table 1 gives the values of the degree of crystallinity of the tested materials. But how were they made? Was the degree of crystallinity determined for the starting materials or for processed materials? From which DSC curve was the degree of criticality determined? Did the material have cold crystallization?

In research goals presented at the end of introduction part, the Authors mention that one of the goals was "…check how the temperature-induced thermal condition inducing crystallization of PLA affected the rate of hydrolytic degradation… " in my work, however, it was completely omitted. That is why it should be completed before publication, which is why I am in favor of a major revision of the article.

Author Response

The answer on Reviewers Comments

We would like to express our grateful for Reviewer for their work, worth remarks and comments. Hereby we would like to answer to him:

From the formal comments regarding the article, I would like to draw attention to the lack of consistency in the naming of samples. Once the samples are described as PLAX e.g. PLA12 and another time as PLAX-YYY e.g. PLA12-DR600.

Thank you for the remark we add omitted information in the section 2.1. Materials and additionally we rewritten sample codes in section 3.1. Photographic documentation and SEM results.

From the substantive comments regarding the article, I would like to draw attention to:

Why is PLA12-DR400 material considered amorphous in the table, and is the crystallinity value already given in PLA12_DR600? Material with a degree of crystallinity of 1.2% would also be considered as amorphous, because with such a small value, the possible measurement error used to determine the degree of crystallinity raises doubts.

However, I have the biggest complaint about the lack of thorough research and description of the methodology of testing the degree of crystallinity of materials.

Table 1 gives the values of the degree of crystallinity of the tested materials. But how were they made? Was the degree of crystallinity determined for the starting materials or for processed materials? From which DSC curve was the degree of criticality determined? Did the material have cold crystallization?

Thank you for remark. We added information about previously paper see ref [35] where we published and structural changes of various PLA during fiber made processing. We well know that the crystallinity around 1-2% is marginally but according to cited paper for the sample PLA12-DR400 the melting point is not observed while in the case of PLA12-DR600 is present. So that we decided to put that information about crystallinity.

What is more we posses the DSC and WAXD results recording during degradation but we planned publish another paper where we focused on the crystallization of fibers materials during degradation.

In research goals presented at the end of introduction part, the Authors mention that one of the goals was "…check how the temperature-induced thermal condition inducing crystallization of PLA affected the rate of hydrolytic degradation… " in my work, however, it was completely omitted. That is why it should be completed before publication, which is why I am in favor of a major revision of the article.

We rewritten that paragraph.

Round 2

Reviewer 3 Report

The authors corrected the errors indicated and answered my questions and doubts. I am in favor of the publication in its current form.